# The Influence of Extra-Fine Milling Protocol on the Internal Fit of CAD/CAM Composite and Ceramic Crowns

**DOI:** 10.3390/ma17225601

**Published:** 2024-11-15

**Authors:** João Paulo Mendes Tribst, Fatema Hosseini, Rafaela Oliveira Pilecco, Carlos Manuel Serrano, Cornelis Johannes Kleverlaan, Amanda Maria de Oliveira Dal Piva

**Affiliations:** 1Department of Reconstructive Oral Care, Academic Centre for Dentistry Amsterdam (ACTA), Universiteit van Amsterdam and Vrije Universiteit, 1081LA Amsterdam, The Netherlands; 2Department of Conservative Dentistry, Faculty of Dentistry, Federal University of Rio Grande Do Sul, Porto Alegre 90035-003, Brazil; rafaela-pilecco@hotmail.com; 3Department of Digital Dentistry, Academic Centre for Dentistry Amsterdam (ACTA), Universiteit van Amsterdam and Vrije Universiteit, 1081LA Amsterdam, The Netherlands; c.serrano@acta.nl; 4Department of Integrated Dentistry & General Oral Care Clinic, Academic Centre for Dentistry Amsterdam (ACTA), Universiteit van Amsterdam and Vrije Universiteit, 1081LA Amsterdam, The Netherlands; 5Department of Dental Materials Science, Academic Centre for Dentistry Amsterdam (ACTA), Universiteit van Amsterdam and Vrije Universiteit, 1081LA Amsterdam, The Netherlands; c.kleverlaan@acta.nl (C.J.K.); a.m.de.oliveira.dal.piva@acta.nl (A.M.d.O.D.P.)

**Keywords:** CAD/CAM, dental materials, dental internal adaptation, milling protocols, resin composite, lithium disilicate

## Abstract

This study aimed to evaluate the marginal and internal adaptation of CAD/CAM crowns milled using two different milling protocols (fine or extra-fine) within a 4-axis milling machine. The crowns were fabricated from lithium disilicate ceramic (IPS e.max CAD) and resin composite (Tetric CAD), assessing their fit in various regions. The crowns (N = 40, n = 10) were milled from lithium disilicate and resin composite using a CEREC Primemill unit. Four groups were formed based on the material and milling protocol: EFLD (extra-fine lithium disilicate), FLD (fine lithium disilicate), EFRC (extra-fine resin composite), and FRC (fine resin composite). The crowns were measured using the replica technique, evaluating internal and marginal adaptation in 18 measuring points per specimen. Data were statistically analyzed using ANOVA and Tukey’s test. Resin composite crowns demonstrated a significantly better internal fit compared to lithium disilicate (*p* < 0.001). Marginal and internal measurements for resin composites were consistently smaller across regions compared to lithium disilicate. No significant differences were found between milling protocols except for the axial wall region (*p* = 0.001), where extra-fine milling resulted in smaller values. Resin composite crowns exhibited superior internal fit compared to lithium disilicate, regardless of milling protocol. Both the fine and extra-fine milling protocols had minimal impact on adaptation, except at the axial wall region, with both protocols promoting adequate results overall.

## 1. Introduction

Digital dentistry, pushed by advancements in computer-aided design and computer-aided manufacturing (CAD/CAM) technologies, has greatly affected the field of prosthetic dentistry. Modern CAD/CAM systems offer an extensive range of materials, including ceramics and resin composites, enabling clinicians to fabricate precise and esthetic dental restorations according to the patient’s needs [1]. Therefore, the digital workflow has improved the fabrication process, enhancing efficiency and patient experience [2].

Despite the significant capabilities of CAD/CAM technology, as with traditional crown techniques, optimal clinical results depend on several material-related, technical, and biological factors. Technical/material complications may result in fractures or loss of retention of the restoration, while biological complications may include caries, endodontic, or periodontal pathology [3]. To address these issues, researchers have extensively investigated the fit of CAD/CAM crowns. Among the methodologies, the replica technique has emerged as a valuable tool allowing for the precise measurement of marginal and internal misfits through accurate examination under microscopy [4]. The replica technique is also recognized for being simple, reproducible, and low-cost when compared to other methods such as microcomputed tomography [4,5].

In addition to the restorative material, other factors can significantly affect the fit of CAD/CAM crowns. The milling protocols, including the type of burs used, play a crucial role in the outcome [6]. Different protocols are available to either increase the milled restoration’s smoothness or reduce the milling time. For example, extra-fine burs are designed to enhance the smoothness and detailing of the restoration’s surface. On the other hand, the fine milling protocol only uses a one-step milling process with conventional burs, which can expedite the milling process, making it more efficient but potentially at the cost of precision [7,8]. Understanding and optimizing these milling parameters is essential for achieving the best possible fit and function of CAD/CAM crowns. Previous studies have demonstrated that milling protocols can affect margin chipping, topographic characteristics, and fracture load of dental restorations [9,10]. However, it is important to state that considering the varying microstructure and machinability of CAD/CAM restorative materials, e.g., glass-ceramics and resin-based materials [10,11], the consequences of milling protocols on the crown’s fit may vary.

Therefore, the purpose of this study was to investigate the internal and marginal fit of CAD/CAM crowns fabricated using different milling protocols and restorative materials. It assessed the internal and marginal fit of resin composite and lithium disilicate ceramic crowns milled with fine and extra-fine milling protocols within a 4-axis CAD/CAM milling system. This study’s hypotheses were that: (i) the crowns fabricated with the extra-fine milling protocol would exhibit a better internal and marginal fit compared to those milled with the fine milling protocol; (ii) and that resin composite crowns would have a better internal and marginal fit compared to lithium disilicate crowns.

## 2. Materials and Methods

This in vitro study is divided into two factors considering: (i) restorative materials—a CAD/CAM resin composite (Tetric CAD, Ivoclar, Ivoclar AG, Schaan, Liechtenstein) and a lithium disilicate ceramic (IPS e.max CAD, Ivoclar AG); (ii) milling protocol—a fine or extra-fine milling strategy. The milled restorations were evaluated considering the marginal and internal fit throughout the replica technique and the topography resultant after milling.

### 2.1. Study Design

Materials used in this study are listed in Table 1. Forty identical glass fiber-reinforced resin epoxy dies were used to replicate a simplified molar crown preparation (n = 10). An individual crown preparation was digitally scanned with an intraoral scanner (CEREC Primescan; Dentsply Sirona, Charlotte, NC, USA), and the resulting three-dimensional images were processed using CAD software (CEREC software 5.2.4, Dentsply Sirona) to design an anatomical crown with an occlusal cement space of 120 µm and a minimal occlusal thickness of 1 mm.

After designing, a 4-axis milling unit (CEREC Primemill, Dentsply Sirona) was used to mill all the crowns. Two different CAD/CAM materials were used: a resin composite (RC; Tetric CAD, Ivoclar AG) and a lithium disilicate glass-ceramic (LD; IPS e.max CAD, Ivoclar AG). The crowns were milled considering the two different milling protocols: fine (-F) using a one-step process with the burs recommended by the manufacturer (Primemill Diamond Burs 1.4 CS and Primemill Diamond Burs 1.2 CS, Dentsply Sirona); or extra-fine (-EF) which uses both fine and extra-fine grinding burs (Primemill Diamond Burs 1.0 mm CS and Primemill Diamond Burs 0.6 mm CS, Dentsply Sirona) in a two-step milling process. The crowns were assigned numerical identifiers based on their sequence of milling. Subsequently, the crowns were detached from their blocks using a dental handpiece with a diamond cutting bur, ensuring structural integrity. Furthermore, the crowns were finished using a polishing bur on the same handpiece, aimed at achieving a smooth surface finish at the location where the crown was attached to the block after the milling protocol. The lithium disilicate crowns were then subjected to crystallization in a furnace (Programat P100, Ivoclar AG) according to the manufacturer’s instructions (closing time: 6 min, temperature gradient 1: 60 °C/min, holding temperature 1: 770 °C, holding time 1: 10 s, holding gradient 2: 30 °C/min, holding temperature 2: 850 °C, holding time 2: 10 min, vacuum 1: 550 until 770 °C, vacuum 2: 770 until 850 °C, long-term cooling: 700 °C/min, and standby temperature: 403 °C). The crowns were randomly assigned in pairs with the glass-fiber-reinforced resin epoxy dies.

### 2.2. Marginal and Internal Fit Evaluation

The marginal fit is defined at the crown margin, while the internal fit was measured at key locations: the cervical-axial angle, axial wall, axio-occlusal angle, and occlusal space (as shown in Figure 1). Each point reflects a distinct region of the crown-tooth interface, allowing for a comprehensive analysis of adaptation across the crown. The replica technique was used to assess the internal and marginal fit of the dental crowns to the die preparation. For this, each crown was first filled with a light body silicone material (3M ESPE Express™, Saint Paul, MN, USA). During the silicone polymerization process, a standardized force of 5N was applied using a calibrated load applicator for 60 s to ensure consistent pressure across all samples. The apparatus was designed to maintain constant pressure and was self-designed by the authors, consisting of a pendulum, lever arm, and load applicator. After allowing the silicone to cure following the manufacturer-recommended duration (5 min), any excess material was carefully removed using a scalpel. Following this, the restoration was carefully removed. A heavy-body silicone (3M ESPE Express™ STD Firmer Set) was applied over the light-body silicone, with a curing time adhering to the manufacturer’s guidelines (5 min).

The heavy body silicone served as a carrier to transfer the light body silicone. Once the light body silicone was within the carrier, the impressions were halved by slicing them in half with a scalpel. After that, the silicone replicas were placed under a stereomicroscope (Zeiss KL2500 LCD, Oberkochen, Germany) under a magnification of 16× with a measuring reference for subsequent analysis. The images were analyzed using inspection software (ImageJ 1.54g, National Institutes of Health, Bethesda, MD, USA). Measurement points on each side of the sliced mold were recorded at 9 distinct locations (margin, cervical-axial angle, axial wall, axio-occlusal angle, and occlusal space on both sides of the impression material), as illustrated in Figure 1, resulting in 18 measurements per crown.

### 2.3. Statistical Analysis

Statistical software was used for all numerical analysis (Minitab 16.1.0, State College, PA, USA) using a significance level of 0.05. Considering both parametric and homoscedastic distribution according to Shapiro–Wilk and Levene’s test, respectively (*p* > 0.05), the marginal and internal data were analyzed descriptively and submitted to a two-way ANOVA test with Tukey’s post hoc test.

## 3. Results

A two-way ANOVA was conducted for each measurement site to assess the impact of material (resin composite vs. lithium disilicate) and milling protocol (fine vs. extra-fine). Significant effects were found for the material at all sites (*p* < 0.05), with resin composite demonstrating a better internal and marginal fit compared to lithium disilicate (Table 2, Table 3, Table 4, Table 5, Table 6 and Table 7). The milling protocol had a significant effect on the axial wall, where the extra-fine milling protocol exhibited superior fit (*p* = 0.043, F = 4.05). Table 2 presents the two-way ANOVA results for marginal fit, indicating a significant difference in fit between resin composite and lithium disilicate materials (*p* < 0.001), with the resin composite consistently displaying a closer fit. In contrast, no significant differences were observed between milling protocols except at the axial wall (Table 4), where extra-fine milling showed a statistically better fit (*p* = 0.001). In Table 2, Table 3, Table 4, Table 5 and Table 6, “Adj SS” stands for “Adjusted Sum of Squares” and “Adj MS” represents “Adjusted Mean Square,” indicating the mean of the adjusted sum of squares.

Table 7 presents the mean values for marginal and internal fit (measured in micrometers) of crowns fabricated from two different restorative materials: resin composite (RC) and lithium disilicate (LD). The fit measurements are divided into specific regions, including the marginal fit, cervical-axial angle, axial wall, axio-occlusal angle, and occlusal space. Resin composite crowns demonstrated a significantly closer marginal fit (mean: 59.5 µm) compared to lithium disilicate crowns (mean: 196.9 µm).

Different letters indicate statistical difference according to two-way ANOVA and Tukey’s post hoc test (α = 0.05).

The interaction between the material and milling protocol was analyzed, showing no significant interaction effects at most sites, indicating that the main effects predominantly drive the outcomes. Whenever significant differences were detected via two-way ANOVA, Tukey’s tests were applied to further explore these differences. A detailed boxplot provided in Figure 2 visualizes the distribution of internal and marginal fit measurements across different groups, illustrating lower variability and tighter interquartile ranges for resin composite crowns. It illustrates the distribution of fit measurements across different groups, with resin composite crowns showing tighter interquartile ranges, suggesting lower variability and more consistent fit compared to lithium disilicate. This graphical representation supports the statistical findings, highlighting the superior internal and marginal fit for crowns milled out of resin composite.

## 4. Discussion

In this study, the milling protocol was shown to significantly affect the crown’s fit at only one site, the axial wall. Therefore, the first hypothesis regarding a significant effect of the milling protocol on the internal and marginal fit was rejected for all parameters except for the preparation space of the axial wall. However, the CAD/CAM restorative material affected the crown’s internal and marginal fit. Thus, the second hypothesis that resin composite crowns would present a better internal and marginal fit than lithium disilicate crowns was accepted.

These findings align with existing literature that highlights the superior adaptability of resin composite materials in CAD/CAM systems [1,12]. Resin composites exhibit a viscoelastic and less hard structure that allows for more precise milling and better adaptation to the prepared tooth structure since it makes material removal during milling easier. Thus, it facilitates a more accurate milling process compared to a more rigid material, such as lithium disilicate ceramics, which frequently present chipping due to machining [10,11,13]. In this sense, our study’s findings align with the existing literature, demonstrating that resin composite crowns exhibit a better internal fit than lithium disilicate, consistent with Rippe et al. (2017) [14]. Moreover, our observation of lower internal discrepancies in resin composites parallels findings by Goujat et al. (2018) [15], who reported a negative correlation between flexural strength and internal discrepancy.

Another factor that could potentially add to the worse fit of lithium disilicate restorations is that this material inherently requires an extra post-milling process. During the crystallization phase, a small volumetric change may occur [16], which poses a challenge in maintaining the internal fit and overall accuracy of the restoration. The crystallization process involves high temperatures to form lithium disilicate crystals from the previous blue stage, which is composed of lithium metasilicate crystals surrounded by silica content. During the restoration design in the CAD software, this volumetric modification, the consequence of crystallization, is usually compensated; however, previous studies have shown that this post-milling procedure in the process of making lithium disilicate crowns inherently adds complexity and potential for error, which is not present in the production of resin composite crowns [17,18]. The impact of the crystallization process on lithium disilicate’s internal fit, observed in our research and corroborated by Alves et al. (2023) [19], highlights the dimensional changes induced by high-temperature processes.

Despite these challenges, the outcomes of our study suggest that the specific properties of resin composites, such as their lower expansion and higher flexibility, make them more suitable for achieving a precise internal fit in CAD/CAM crowns. The extra-fine milling protocol was expected to produce a smoother surface finish and better fit; however, it did not demonstrate a significant impact, except at the axial wall. This finding may suggest that while a second milling protocol with finer burs can improve surface smoothness, their effect on overall fit may be less significant than initially hypothesized. It is important to state that our use of the replica technique to measure internal fit is supported by Ferrairo et al. (2021) [20], validating its accuracy across different CAD/CAM systems. Notably, across all studies, including ours, all materials and CAD/CAM systems yielded clinically acceptable crowns, highlighting the efficacy of current digital dentistry technologies in producing reliable dental restorations [14,15,17,19,20].

In addition, the limited effect of milling protocols on overall adaptation, aside from the axial wall, may be attributed to the uniform surface geometry of other regions, which are less affected by fine details achieved in extra-fine milling. At the axial wall, where the material’s rigidity and the milling tool’s finer detailing capabilities interact, extra-fine milling enables a more precise fit, suggesting the protocol’s role in achieving optimal adaptation in wider areas from the intaglio surface. It can be suggested that, because the axial wall generally has an extended surface area in contact with the tooth structure compared to other regions, any minor inaccuracies are more likely to affect the overall fit.

The average fit of the composite crowns in this study was measured at 137.8 µm, consistent with values reported in previous research investigating the marginal and internal fit of nanocomposite CAD/CAM restorations. In a previous study, the fit values were 149.76 ± 58.36 µm for an experimental resin block and 120.82 ± 46.72 µm for Lava Ultimate [21]. These findings are in close agreement with the present results, indicating that the fit of composite crowns falls within a similar range across different materials and studies.

A previous study aimed to evaluate the marginal and internal adaptation of CAD/CAM ceramic and composite resin crowns with varying internal spacings using microcomputed tomography [22]. The findings demonstrated that the adjustment and spacing factors were not significantly different in their interaction for adaptation scores (AS). Only the material type showed a statistically significant effect on AS (*p* < 0.001), with ceramics presenting significantly lower AS values compared to composite resin (ANOVA 1-way; *p* < 0.001). Specifically, the ceramic crowns exhibited a mean ± SD of 79 ± 17 μm, while the composite resin crowns had a mean of 101 ± 21 μm. In contrast, the average adaptation score for lithium disilicate crowns in the present study was 210.2 μm. This discrepancy may be attributed to differences in the seating load; the previous study applied a 19.6 N load for 3 min to extrude excess material, while our study used only a 5 N load. Additionally, our study employed a minimum cement space setting of 120 μm. In contrast, the other research utilized tighter settings of 80 or 30 μm, which may further account for the observed variation in misfit.

It is important to recognize the limitations of comparing two materials with distinct properties. However, since they share the same clinical indications (veneers, inlays, onlays, and crowns), the comparison is still relevant for clinicians aiming to choose the most predictable outcome for treating patients. Lithium disilicate ceramics are known for their high strength and excellent esthetic qualities. The crystalline structure provides a high level of durability, making them ideal for areas that require significant load-bearing capacity. However, the same crystalline structure that gives lithium disilicate its strength also makes it less flexible, which requires careful handling and precise milling techniques to avoid damage during the manufacturing process. Supporting that, a previous study demonstrates that optimizing soft machining processes (minimizing machining forces and surface roughness) aids in achieving both acceptable surface and edge quality while maintaining balanced removal rates [23]. Resin composites, being polymer-based materials, offer greater flexibility. The polymer matrix within these composites can absorb and distribute stress more evenly, which is advantageous during both the fabrication and placement of restorations. This stress absorption and distribution occur through the viscoelastic behavior of the polymer matrix but are also influenced by other factors such as time, temperature, and curing methods. Additionally, efficient packing and proper gradation of reinforcing particles further enhance stress distribution [24,25]. In this study, Tetric CAD (Ivoclar Vivadent, Liechtenstein), an esthetic composite, was utilized, being composed of Bis-GMA, Bis-EMA, TEGDMA, UDMA nano-filled 70% with barium glass, and silicon dioxide. According to the data from the literature, the bending resistance is 273.8 MPa, with an elastic modulus of 10.2 GPa. This elastic modulus value, closer to that of dentine (18 GPa), is suggested to reduce the concentration of stress in the restoration and dampen fracture chances [24,25]. Clinically, the choice between lithium disilicate ceramics and resin composites depends on several factors, including the location of the restoration, the patient’s bite force, esthetic requirements, and the specific clinical situation [13,25]. In summary, innovations in materials and manufacturing processes, such as the extra-fine milling process, can lead to more cost-effective solutions for patients, especially when restorative material selection can be carried out as a patient-centered approach when making dental treatment decisions [26].

The implications of the results in this study are significant for clinical practice. It suggests that while the choice of material plays a crucial role in the fit of CAD/CAM crowns, the milling settings, particularly the milling protocol used, might not be as critical as previously thought for most internal surfaces. However, for large areas such as the axial wall, a second step with finer burs might still offer some advantages. Further research is warranted to explore the impact of other milling parameters and to validate these findings across different types of CAD/CAM systems and restorative materials. In addition, different luting cements can also behave differently in the final cement space due to their viscosity and mechanical properties [25]. Clinicians should also consider these factors when selecting materials and milling protocols for CAD/CAM restorations to optimize fit and longevity.

## 5. Conclusions

The study confirms that material selection significantly impacts the fit of CAD/CAM crowns, with resin composites showing superior adaptability compared to lithium disilicate. Although milling protocol effects were minimal, extra-fine milling demonstrated improved adaptation at the axial wall, highlighting its potential role in refining fit in some regions of the crown.

Further research should explore the interaction between milling parameters and materials to refine CAD/CAM protocols and restorative material selection.

## Figures and Tables

**Figure 1 materials-17-05601-f001:**
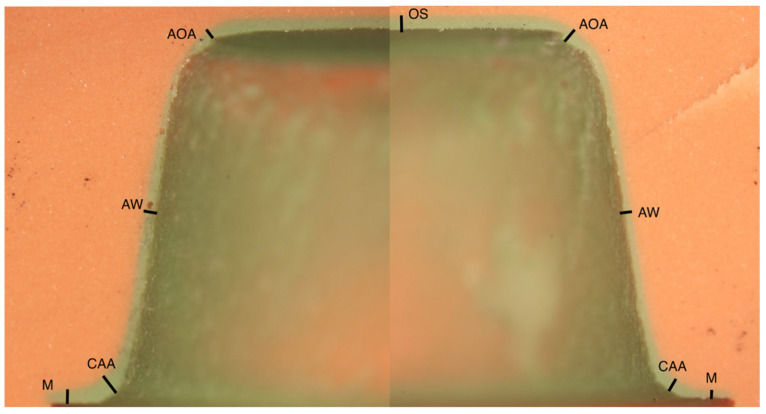
Different measuring points on the sectioned impression: margin (M), cervical-axial angle (CAA), axial wall (AW), axio-occlusal angle (AOA), occlusal space (OS).

**Figure 2 materials-17-05601-f002:**
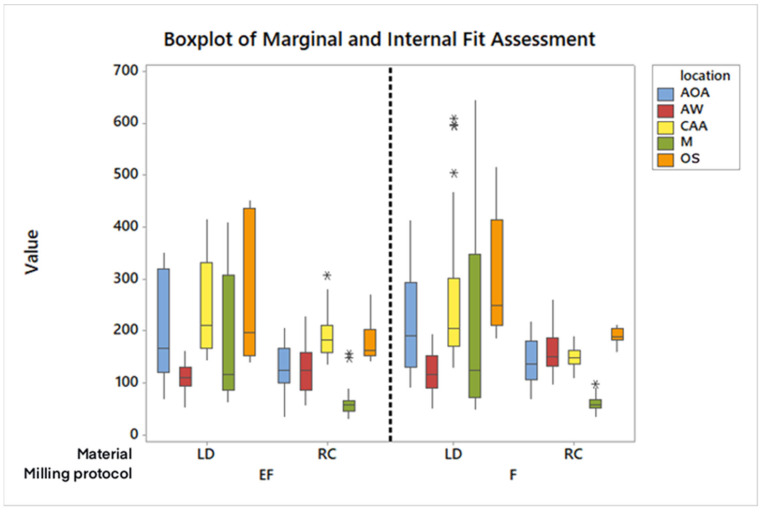
Boxplot with internal and marginal crown preparation space in µm on the *y*-axis and material and milling protocol on the *x*-axis. The dashed line indicates the separation between extra-fine and fine milling protocols. Asterisks represent data values that are far away from others.

**Table 1 materials-17-05601-t001:** Materials description and commercial names.

Material	Commercial Name
CAD/CAM resin composite	Tetric^®^ CAD, Ivoclar AG, Schaan, Liechtenstein
CAD/CAM lithium disilicate	IPS e.max^®^ CAD, Ivoclar AG
Intraoral Scanner	CEREC Primescan; Dentsply Sirona, Charlotte, NC, USA
CAD software	CEREC software 5.2.4, Dentsply Sirona
CAM milling machine	CEREC Primemill, Dentsply Sirona
Diamond milling burs used for CAD/CAM technology—fine milling protocol	Diamond 1.2 CS (6714070), Dentsply SironaDiamond 1.4 CS (6714088), Dentsply Sirona
Diamond milling burs used for CAD/CAM technology—extra-fine milling protocol	Diamond 1.2 CS (6714070), Dentsply SironaDiamond 1.4 CS (6714088), Dentsply SironaDiamond 0.6 CS (6714054), Dentsply SironaDiamond 1.0 CS (6714062), Dentsply Sirona
Glass fiber-reinforced epoxy resin die	Protec Produtos Técnicos Ltd.a., São Paulo, Brazil
Light body silicone	3M ESPE Express™, Saint Paul, MN, USA
Heavy body addition silicone material	3M ESPE Express™

**Table 2 materials-17-05601-t002:** Two-way ANOVA result for marginal fit.

Source	DF	Adj SS	Adj MS	F-Value	*p*-Value
Milling protocol	1	9226	9226	0.7	0.404
Material	1	755,013	755,013	57.27	<0.001
Milling protocol × Material	1	10,256	10,256	0.78	0.379
Error	156	2,056,503	13,183		
Total	159	2,830,998			

**Table 3 materials-17-05601-t003:** Two-way ANOVA result for cervical-axial angle.

Source	DF	Adj SS	Adj MS	F-Value	*p*-Value
Milling protocol	1	8836	8836	1.21	0.273
Material	1	271,673	271,673	37.17	<0.001
Milling protocol × Material	1	27,431	27,431	3.75	0.055
Error	156	1,140,205	7309		
Total	159	1,448,145			

**Table 4 materials-17-05601-t004:** Two-way ANOVA result for axial wall.

Source	DF	Adj SS	Adj MS	F-Value	*p*-Value
Milling protocol	1	17,119	17,119	11.38	0.001
Material	1	26,910	26,910	17.89	<0.001
Milling protocol × Material	1	5256	5256	3.49	0.063
Error	156	234,619	1504		
Total	159	283,904			

**Table 5 materials-17-05601-t005:** Two-way ANOVA result for axio-occlusal angle.

Source	DF	Adj SS	Adj MS	F-Value	*p*-Value
Milling protocol	1	8791	8791	1.53	0.218
Material	1	215,502	215,502	37.44	<0.001
Milling protocol × Material	1	60	60	0.01	0.919
Error	156	898,027	5757		
Total	159	1,122,381			

**Table 6 materials-17-05601-t006:** Two-way ANOVA result for occlusal space.

Source	DF	Adj SS	Adj MS	F-Value	*p*-Value
Milling protocol	1	12,802	12,802	1.76	0.188
Material	1	185,281	185,281	25.53	<0.001
Milling protocol × Material	1	1824	1824	0.25	0.618
Error	76	551,564	7257		
Total	79	751,471			

**Table 7 materials-17-05601-t007:** Post hoc result of the internal and marginal crown preparation space in µm for the different restorative materials (N = measured points).

Location	Material	N	Mean (µm)	Grouping	
Marginal Fit	LD	80	196.9	A	
	RC	80	59.5		B
Cervical-Axial Angle	LD	80	251.8	A	
	RC	80	169.4		B
Axial Wall	RC	80	142.4	A	
	LD	80	116.4		B
Axio-Occlusal Angle	LD	80	209.8	A	
	RC	80	136.4		B
Occlusal space	LD	40	279.4	A	
	RC	40	183.1		B

Different letters indicate statistical difference according to two-way ANOVA and Tukey’s post hoc test (α = 0.05).

## Data Availability

The original contributions presented in the study are included in the article, further inquiries can be directed to the corresponding author.

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
