# Peer review of "The Influence of Extra-Fine Milling Protocol on the Internal Fit of CAD/CAM Composite and Ceramic Crowns"

_materials, 2024, doi:10.3390/ma17225601_

Round 1
Reviewer 1 Report
Comments and Suggestions for Authors
This article reported a study of the effect of milling protocol on the internal fit of CAD-CAM composite and ceramic crowns. The paper needs a substantial revision before it can be accepted. For example:
1. Abstract
The abstract should be revised. There is no need to divide the abstract into different subsections.
2. Results
The explanation of Tables 2-7 and Figure 2 is poor, and a substantial revision is needed.
3. Discussion
The authors showed only two summaries in the Conclusion section: 1) material selection significantly impacted the internal and marginal fit paper shows, which is not new for readers; and 2) Both milling protocols had minimal impact on adaptation, except at the axial wall region.
Therefore, the reasons why the milling protocols did not demonstrate a significant impact, except at the axial wall should be well discussed, and the significance of this finding should be addressed.
Author Response
This article reported a study of the effect of milling protocol on the internal fit of CAD-CAM composite and ceramic crowns. The paper needs a substantial revision before it can be accepted. For example:
Dear reviewer, thank you for your time and effort in evaluating the present study. Your comments were significant to improve the manuscript.
- Abstract
The abstract should be revised. There is no need to divide the abstract into different subsections.
The abstract has been reviewed as suggested.
- Results
The explanation of Tables 2-7 and Figure 2 is poor, and a substantial revision is needed.
The tables and figure explanation have been improved. However, most of the tables in this study are self-explanatory.
- Discussion
The authors showed only two summaries in the Conclusion section: 1) material selection significantly impacted the internal and marginal fit paper shows, which is not new for readers; and 2) Both milling protocols had minimal impact on adaptation, except at the axial wall region.
Therefore, the reasons why the milling protocols did not demonstrate a significant impact, except at the axial wall should be well discussed, and the significance of this finding should be addressed.
The discussion section has been expanded.
Reviewer 2 Report
Comments and Suggestions for Authors
I think this research was conducted using relatively appropriate research methods and procedures.
It appears that there are several areas for confirmation and supplementation.
1) Specific descriptions are required for the 5N force and 60 seconds time in the silicone polymerization process. For example, specific descriptions of the equipment or tools used are required.
2) More specific definitions of the marginal and internal fit measurement points are required.
3) It's believed that if the method of presenting the results of the two-way ANOVA for each measurement points is simplified, the paper will be more readable.
Author Response
I think this research was conducted using relatively appropriate research methods and procedures.
It appears that there are several areas for confirmation and supplementation.
Dear reviewer, thank you for taking the time to review this study. Your feedback really helped improve the manuscript.
1) Specific descriptions are required for the 5N force and 60 seconds time in the silicone polymerization process. For example, specific descriptions of the equipment or tools used are required.
The methods section has been improved with this info.
2) More specific definitions of the marginal and internal fit measurement points are required.
The methods section has been improved as suggested.
3) It's believed that if the method of presenting the results of the two-way ANOVA for each measurement points is simplified, the paper will be more readable.
We agree. However, since the tables are self-explanatory and contribute to open-science information, we prefer to keep them.
Reviewer 3 Report
Comments and Suggestions for Authors
The paper considers the production by cad/cam manufacturing of dental reparations from two different materials, one being ceramic in structure LiSilicate and the other being polymeric one. The methods used to produce the part were based on the milling process, which is a classical part that produces very sophisticated parts that should lead to quasi-perfect parts.
The authors reported that the polymeric material is better in dimensional tolerance compared to the ceramic one. The authors don't give a good explanation for this. There is some hypothesis considering the structure of the material, but those are not sufficiently well considered, and the reasons are not well explained.
There is no structural characterization of materials used in the paper. All the information the authors gave about the materials is their vague chemical compositions, but no real comparison of the two is given.
The topic is important but needs to be adapted for publication in the Materials Journal.
I suggest a text revision that would include a detailed review of the materials' structure and a comparative analysis of the structures and surfaces of produced parts.
Author Response
The paper considers the production by cad/cam manufacturing of dental reparations from two different materials, one being ceramic in structure LiSilicate and the other being polymeric one. The methods used to produce the part were based on the milling process, which is a classical part that produces very sophisticated parts that should lead to quasi-perfect parts.
Dear Reviewer, thank you for your time in suggesting improvements for this study. We appreciate.
The authors reported that the polymeric material is better in dimensional tolerance compared to the ceramic one. The authors don't give a good explanation for this. There is some hypothesis considering the structure of the material, but those are not sufficiently well considered, and the reasons are not well explained.
There is no structural characterization of materials used in the paper. All the information the authors gave about the materials is their vague chemical compositions, but no real comparison of the two is given.
The topic is important but needs to be adapted for publication in the Materials Journal.
I suggest a text revision that would include a detailed review of the materials' structure and a comparative analysis of the structures and surfaces of produced parts.
The text has been extended with this information. The discussion was added to include more aspects of the differences in physical properties that affect their machinability and fit.
Round 2
Reviewer 1 Report
Comments and Suggestions for Authors
To facilitate readers’ understanding, it is better to provide under Table 2 an explanation of the abbreviations (e.g., Adj SS, Adj MS) in the table. In addition, a further explanation of Table 7 is suggested.
Author Response
Response to Reviewer:
Thank you for the suggestions to enhance the clarity of our manuscript.
Modifications Made:
-
Explanation of Abbreviations in Table 2: We have added an explanation below Table 2 for abbreviations such as "Adj SS" (Adjusted Sum of Squares) and "Adj MS" (Adjusted Mean Square), to help readers better understand the statistical terms used in the analysis.
-
Additional Explanation for Table 7: We included an explanatory section for Table 7.
Reviewer 3 Report
Comments and Suggestions for Authors
No further comments
Author Response
Comments and Suggestions for Authors: No further comments Reply: Thank you.